# Magnitude, pattern and correlates of multimorbidity among patients attending chronic outpatient medical care in Bahir Dar, northwest Ethiopia: The application of latent class analysis model

**Fantu Abebe Eyowas**[1,2]*, **Marguerite Schneider**[3], **Shitaye Alemu**[4], **Sanghamitra Pati**[5], **Fentie Ambaw Getahun**[1]

1 School of Public Health, College of Medicine and Health Sciences, Bahir Dar University, Bahir Dar, Ethiopia, 2 Jhpiego corporation, Bahir Dar Regional Office, Bahir Dar, Ethiopia, 3 Department of Psychiatry and Mental Health, University of Cape Town, Cape Town, South Africa, 4 School of Medicine, College of Medicine and Health Sciences, University of Gondar, Gondar, Ethiopia, 5 ICMR-Regional Medical Research Center, Bhubaneswar, India

* fantuabebe@gmail.com

**Data Availability Statement:** I used Harvard dataverse platform to publish the data. Please find the information (DOI) below. Abebe, Fantu, 2022,

## Abstract

### Objective

This study aimed to investigate the magnitude, pattern and associated factors of multimorbidity in Bahir Dar, northwest Ethiopia.

### Methods

A multi-centered facility-based study was conducted among 1440 participants aged 40+ years attending chronic outpatient medical care. Two complementary methods (interview and review of medical records) were employed to collect data on socio-demographic, behavioral and disease related characteristics. The data were analyzed by STATA V.16 and R Software V.4.1.0. We fitted logistic regression and latent class analyses (LCA) models to identify the factors associated with multimorbidity and determine patterns of disease clustering, respectively. Statistical significance was considered at P-value <0.05.

### Results

The magnitude of individual chronic conditions ranged from 1.4% (cancer) to 37.9% (hypertension), and multimorbidity was identified in 54.8% (95% CI = 52.2%-57.4%) of the sample. The likelihood of having multimorbidity was higher among participants aged 45–54 years (AOR: 1.6, 95%CI = 1.1, 2.2), 55–64 years (AOR: 2.6, 95%CI = 1.9, 3.6) and 65+ years (AOR: 2.6, 95%CI = 1.9, 3.6) compared to those aged 40–44 years. The odds of multimorbidity was also higher among individuals classified as overweight (AOR: 1.6, 95%CI = 1.2, 2.1) or obese (AOR: 1.9, 95%CI = 1.3, 3.0) than the normal weight category. Four patterns of multimorbidity were identified; the cardiovascular category being the largest class

"Multimorbidity of chronic conditions in Bahir Dar, Ethiopia, 2022", https://doi.org/10.7910/DVN/B06WPU, Harvard Dataverse, V1, UNF:6:Ax/i1nFLjiEP1nwrGx3Sbg== [fileUNF].

**Funding:** Bahir Dar University has partially funded this study. FG also received support from AMARI (African Mental Health Research Initiative), which is funded through the DELTAS Africa Initiative (DEL-15-01) to pursue his studies previously.

**Competing interests:** The authors have declared that no competing interests exist.

(50.2%) followed by the cardio-mental, (32.6%), metabolic (11.5%) and respiratory (5.7%) groups. Advanced age, being overweight and obesity predicted latent class membership, adjusting for relevant confounding factors.

## Conclusions

The magnitude of multimorbidity in this study was high, and the most prevalent conditions shaped the patterns of multimorbidity. Advanced age, being overweight and obesity were the factors correlated with multimorbidity. Further research is required to better understand the burden of multimorbidity and related factors in the population, and to determine the impact of multimorbidity on individuals' well-being and functioning.

## Introduction

Globally, one in four adults are living with at least one chronic condition, of which 80% have two or more long lasting and incurable non-communicable diseases (NCDs), commonly known as multimorbidity [1]. Multimorbidity is a relatively new concept in research globally [2] and there remains a huge interest in answering what is philosophically adequate to define the concept [3] and measure its epidemiology [4, 5].

Debates remain on what constitute chronic conditions and how many chronic conditions to consider towards the measurement of multimorbidity [6, 7]. This challenge led researchers to employ quite diverse methodologies to study the epidemiology of multimorbidity in different contexts [7, 8]

Inconsistencies were mainly observed in terms of the age group of the population studied, the types and number of chronic NCDs considered, study setting, method of data collection and sample size, among others [9, 10]. The use of different methodologies resulted in a huge variation in the prevalence estimates and difficulty in comparing and pooling results of multimorbidity studies globally [6, 11].

Furthermore, although the need to understand the patterns of disease combinations/clusters and associated complexity and care is also well recognized [12], the available literature lacks consistency in the methodologies used to determine the patterns of multimorbidity and the factors underlying these [13].

Despite the inconsistency in the methodologies employed to define and measure multimorbidity, evidence has shown that the burden of multimorbidity is growing globally [2, 14]. Highest prevalence rates were reported from studies in high income countries, where about one in four adults experience multimorbidity [15, 16]. A few studies in low-and middle-income countries (LMICs) have also reported an increasing trend in the burden of NCDs multimorbidity [2, 14, 17].

The prevalence of multimorbidity increases substantially with age [6]. However, the absolute number of people with multimorbidity has also been found to be higher in those younger than 65 years [1, 10]. Multimorbidity is also associated with socioeconomic deprivation and evidence has shown that those living in the most deprived areas develop multimorbidity 10 to 15 years earlier than in the least deprived decile of the population and tend to have multimorbidity accompanying mental health problems [1]. Female gender and obesity are also common risk factors for the occurrence of multimorbidity globally [9].

Despite the huge challenge of multimorbidity, there is a significant knowledge gap in terms of the burden, associated risk factors, and the pattern and distribution of multimorbidity in

Ethiopia. If health systems are to meet the needs and priorities of individuals living with multi-morbidity in the country, we must first accurately measure the burden and distribution of NCDs multimorbidity and associated factors in the context.

This study is part of an ongoing longitudinal study being conducted in a broader range of health care facilities providing health care for the people living with NCDs in the region. This study aimed to determine the magnitude and patterns of multimorbidity and associated factors among individuals attending chronic outpatient medical care in public and private health facilities in Bahir Dar city, Ethiopia.

## Methods and materials

This study presents the first part of an ongoing research project. The detail of the methods employed for the whole research project has been published elsewhere [18], and those methods applied in the current study are summarized below.

### Study design and settings

This is a multi-center facility based cross-sectional study conducted in Bahir Dar City, Ethiopia. The city is the capital of Amhara region–the second most populous region in the country where about 30 million people are living.

Five hospitals (three public and two private) and three private higher/specialty clinics were enrolled in this study. These facilities provide the bulk (~80%) of chronic NCDs care for the people living in the city and surrounding areas. Chronic NCDs care and management is presumed to be provided in a relatively uniform fashion using the national NCDs treatment guideline [19]. However, the nature of patients vising these facilities may vary and there remains a concern on quality and affordability NCDs care in public hospitals and private health facilities, respectively.

A two-stage clustered stratified random sampling method was applied for recruiting the facilities and study participants. The sample size from each facility was determined based on the notion of probability proportional to size (PPS) using the pool of chronic NCD patients ($\geq$ 40yrs) registered for follow-up over the year preceding our assessment (January–December 2020) in each participating facility.

Only facilities who were providing chronic NCDs care by general practitioners or specialist physicians for at least one year's duration prior to the data collection period were considered. Older adults (40 years or more) diagnosed with at least one NCD and on chronic diseases follow up care for at least six months prior to the study period were recruited for the study. Pregnant women and individuals who are too ill to be interviewed and admitted patients were excluded. A sample of 1440 adult patients ($\geq$40yrs) attending chronic outpatient medical care in the selected health facilities were randomly selected and enrolled from March 15 to April 30, 2021 for this baseline study.

### Sample size

The minimum sample size required was determined based on the methods described in our published protocol [18]. Key issues considered to estimate the sample size required were the nature of the dependent variable/s, predictor variables, study designs and analysis techniques. The input values; $\alpha$ (type I error = 0.05), power (1-$\beta$ = 90), confidence level (95%) and the estimated non-response and attrition during follow-up (20%) remain constant when using different formulae. We found the general linear multivariate model with Gaussian errors (GLIMMPSE) sample size and power calculator [20–22] as an appropriate method to yield the

maximum sample size required for the current and future studies using simulated inputs compared to the sample size calculated based on other common methods.

Based on the given assumptions and the formula we used, the sample size required was 600. As the nature of participants is likely to be different by the type of facility (public or private) where they receive care, we employed stratification to ensure fair representation in the sample for important sub-groups. Hence, a design effect of 2 was considered to avoid possible loss of sample during stratification. Adding 20% to the possible loss to follow-up and nonresponse, the sample size needed was calculated to be 1440.

## Definition and measurement of multimorbidity

To facilitate comparison, multimorbidity was operationalized as the co-occurrence of two or more of the chronic NCDs specified below. List of NCDs considered in this study were determined based on a review study [18], and includes hypertension, diabetes, heart disease (heart failure, angina and heart attack), stroke, bronchial asthma, chronic obstructive pulmonary diseases (COPD), depression and cancer.

Moreover, based on a pilot study we conducted, other prevalent chronic diseases (accounting for 2–5%), including musculoskeletal disorders (arthritis, chronic back pain and osteoporosis), thyroid disorders (hyperthyroidism and multinodular goiter), chronic kidney disease, gastrointestinal disorders (chronic liver, gall bladder and gastric diseases) and Parkinson's disease (PD) were identified and included in our assessment. Information on these chronic conditions was assessed through a question asking if the person had ever been diagnosed with the disease by a health professional. The specific question was, "Have you ever been told by a health professional/doctor that you have (disease name)?" responses were either yes (scored as "1") or no (score as "0"). Participants were also prompted to report up to three additional chronic conditions they are living with if any.

To complement the data obtained from interviews [23, 24], we reviewed medical records of all the study participants for recorded medical diagnoses. Except for mental health status (depression), direct assessment of chronic diseases was not possible due to methodological and logistics challenges. We used a locally validated patient health questionnaire (PHQ-9) to measure mental health status of the study participants [25]. Possible PHQ-9 scores range from 0 to 27 and patients scoring 10 or more were classified as having depression [18].

## Measurement of independent variables

Independent variables including socio-demographic characteristics [age, sex, education, marital status, residence and occupation] were measured by a face-to-face interview using pretested and standardized tools. We used the WHO's STEPs survey tool [26] which has been validated and used in Ethiopia to assess dietary habits [amount and frequency of fruit and vegetables consumption and amount of daily salt consumption], and behavioral and lifestyle patterns [alcohol consumption, smoking, Khat consumption and physical exercise]. Data to calculate body mass index (BMI) and waist and hip circumference were directly measured from patients according to the approaches described below.

Height and weight was measured using standardized techniques with participants barefoot and wearing light clothing. Participants height was measured to the nearest 0.1 cm using a portable Seca 213 Stadiometer and weight was recorded to the nearest 0.1 kg using a weighing scale. These data were used to calculate individual body mass index (BMI; kg/m²). BMI values are classified into categories for each individual based on established WHO cut-offs for BMI, which included four categories: underweight ($<$18.5 kg/m²), normal (18.5–24.9 kg/m²), overweight (25.0–29.9 kg/m²), and obese (30 kg/m²) [27]. A flexible, stretch-resistant tape was used

to measure waist and hip circumference to the nearest 0.1 cm midway between the 12[th] rib and the iliac crest and around the widest portion of the hips, respectively. Participants' waist-to-hip ratio (WHR) was estimated and interpreted according to the WHO's protocol [28].

Wealth Index (a latent construct) at household level was generated from a combination of material assets and housing characteristics according to the demography and health survey guideline used in Ethiopia [29]. The Wealth index was scored using a principal component analysis (PCA) technique. The scores were classified into quintiles, differently for urban and rural residents separately, while quintile 1 represents the poorest and quintile 5 the wealthiest [30]. It was collapsed into three classes as low, middle and high income for simplifying its interpretability.

## Data collection procedures

After obtaining consent from the participants, face-to-face interviews were conducted to collect data on socio-demographic characteristics, dietary practices, lifestyle habits and doctor diagnosed medical conditions. Then, measurements of weight, height and waist circumference were taken. Finally, patient charts (medical records) were reviewed to capture recorded medical diagnoses, medications prescribed, latest blood pressure values, fasting blood glucose (FBG) level, and glycated hemoglobin (HbA1c) values and HIV status if available. In addition, data on COVID-19 infection were gathered at different points in time through patient interviews and review of medical records.

Physicians and nurses working in the chronic care unit were involved in the data collection process. However, data were primarily collected by ten graduate nurses recruited from institutions outside the study facilities.

For the sake of a more efficient and accurate data collection, aggregation and statistical analysis, the data were collected by the Kobo Toolbox software [31]. Patients were interviewed and assessed following their usual consultation periods.

## Data quality assurance

Data were collected from multiple sources using pilot tested and standardized instruments. That helped us to validate and rectify congruence between the data captured from different sources (interview and review of medical records). If the data on NCDs was inconsistent between the two sources, the data obtained from medical records was considered more reliable. The use of the Kobo toolbox software helped to collect real time data and monitor the validity of the information entered daily from all the data collectors [31]. The questionnaires to collect the data were translated into Amharic (local language) and pilot tested for cross-cultural adaptability based on standard protocols [32, 33]. Data collectors and supervisors received a two-day training detailing the study, including obtaining written consent, conducting face-to-face interview, performing physical measurement, medical record review and navigating through the questionnaires in the Kobo toolbox platform preloaded onto their smart phones. The data collection process was monitored by trained supervisors and the principal investigator and the data sent every day to the Kobo toolbox server were checked for completeness, accuracy and clarity.

## Data analysis

The data from the Kobo toolbox were downloaded into an Excel spreadsheet and exported to SPSS V. 21 for cleaning and finally analyzed by STATA V. 16 and R Software V.4.1.0. Descriptive statistics were computed to describe the sociodemographic, lifestyle and other characteristics of participants. The magnitude of individual chronic conditions was determined and Chi-squared test run to determine the difference in distribution of individual NCDs between men

and women. The prevalence of multimorbidity was determined by combining evidence mainly from two sources, patient interviews and medical record reviews. The dependent variable (multimorbidity) was coded as yes or no.

The association between the dependent variable (multimorbidity) and independent variables was assessed by fitting a bivariate logistic regression model. The unadjusted odds ratio (OR) with 95% confidence intervals and p-values are reported for each of the independent variables analyzed. Variables having a p-value <0.2 were fitted into the multivariable logistic regression model to predict the adjusted effect of the independent variables on multimorbidity. Before running the multivariable analysis multi-collinearity between independent variables was checked using the Variance Inflation Factor (VIF), and variables were found to be not strongly correlated (the highest value was 2.11). A p–value < 0.05 was taken as statistically significant.

Latent class analysis (LCA) [34] was fitted into R 4.1.0 software (poLCA package) to identify the subgroups of patients sharing class membership. LCA is a powerful method for identifying unobserved groups with categorical outcome variables such as patterns of multimorbidity in a given population [35]. Identifying subgroups of individuals could help in designing appropriate interventions for managing multimorbidity [36]. Given that there is no single indicator reflecting an optimal model fit, model selection was based on a balance of parsimony, substantive consideration and several fit indices [37].

We used the Akaike Information Criterion (AIC) and the Bayesian Information Criterion (BIC) to evaluate the relative fit and adequacy of the models. Lower values on the AIC and the BIC indicate a better-fitting model [37]. The BIC tends to select simpler models than the AIC, and in a Monte Carlo simulation it has been shown to be the most reliable criteria when deciding on the optimal latent class model [38].

Predictors of class membership were identified by running multinomial logistic regression analysis. All the assumptions for multivariable and multinomial logistic regression analyses were checked and the fit statistics are reported accordingly. Key parameter estimates for fitting an adequate LCA model were also assessed and reported accordingly.

## Results

### Characteristics of the study participants

Of the total 1440 study participants enrolled, complete data were obtained from 1432 individuals giving rise to a response rate of 99.4%. The majority (n = 1007, 70.3%) of study participants were urban residents and females constitute a slightly higher (51%) percentage in terms of sex distribution. The mean age of the participants was 56.4 (ranged from 40–93 years) with a standard deviation (SD) of 11.8 years. Individuals aged 45–54 years and 55–64 years accounted equally (27.9) for the age distribution and those aged 65+ a 26.9% share of the total sample. The majority of participants (n = 1081, 75.5%) were married at the time of data collection. For education level, the majority (n = 780, 54.5%) did not have any formal education, while 11.6%, 11.9% and 22% of the participants had achieved primary, secondary and college level or above education programs, respectively. Housewives (23%) and employed individuals (22.9%) represent the largest share in the occupation category. The highest percentage (n = 536, 37.4%) of the participants had a low socioeconomic status (SES), and 28.3% (n = 405) and 34.3% (n = 491) of them fell on the middle and highest SES category, respectively (Table 1).

### Lifestyle, behavioral and psychosocial characteristics

Of all participants, 37% (n = 530) reported that they were doing moderate/high intensity exercise at least three times per week. With regard to dietary habits, 39.3% (n = 563) and 51.5%

**Table 1. Socio-demographic characteristics of study participants attending chronic outpatient NCDs care in Bahir Dar, Ethiopia (N = 1432).**

| Variables | Frequency | Percentage |
|---|---|---|
| Age ≤44Yrs | 247 | 17.3 |
| 45-54Yrs | 399 | 27.9 |
| 55-64Yrs | 400 | 27.9 |
| 65+Yrs | 386 | 26.9 |
| Sex Male | 702 | 49.0 |
| Female | 730 | 51.0 |
| Marital status Currently married | 1081 | 75.5 |
| Single* | 351 | 24.5 |
| Education No formal education | 780 | 54.5 |
| Primary | 166 | 11.6 |
| Secondary | 171 | 11.9 |
| College level and above | 315 | 22.0 |
| Occupation Housewife | 329 | 23.0 |
| Employed | 328 | 22.9 |
| Farmer | 288 | 20.1 |
| Trader | 207 | 14.5 |
| Retired | 141 | 9.8 |
| Unemployed | 139 | 9.7 |
| Wealth Index (SES) Poorest | 269 | 18.8 |
| Poorer | 334 | 23.3 |
| Middle | 267 | 18.6 |
| Rich | 252 | 17.6 |
| Richest | 310 | 21.6 |

*Includes never married, divorced, widowed and separated

(n = 737) of them reported to having consumed fruit and vegetables three times or more per week, respectively. About one sixth (n = 220, 15.4%) of them reported having drunk alcohol of any kind in the past one week prior to the interview and 89 (6.2%) of the participants reported a past history of smoking. However, ascertaining whether the dietary, lifestyle and behavioral practices were happening before the onset of chronic conditions was difficult owing to the risk of recall bias.

The highest percentage (n = 763, 53.3%) of participants had normal body mass index (BMI), whereas overweight and obesity together accounted for 32%. However, based on the waist-to-hip ratio health risk classification, the highest proportion (n = 810, 56.6%) were on the high health risk category (Table 2).

## Magnitude of individual NCDs and number of chronic NCDs per person

The magnitude of each of the chronic conditions considered in this study is shown in Fig 1. The number of NCDs identified per person ranged from one to four (mean = 1.74, SD = 0.78). Hypertension was the most frequently identified NCD (63.5%), followed by diabetes (42.5%) and heart disease (25.6%).

Apparent gender differences were observed in the distribution of individual NCDs, including diabetes, stroke, depression and thyroid disorders. While females were more vulnerable to depression and hyperthyroidism, males tend to get more affected by diabetes, stroke and musculoskeletal disorders (Table 3).

**Table 2. Lifestyle, behavioral and psychosocial characteristics, of participants attending chronic outpatient NCDs care in Bahir Dar, Ethiopia (N = 1432).**

| Variables | Frequency | Percentage |
|---|---|---|
| Moderate/High intensity physical exercise | | |
| Yes | 530 | 37.0 |
| No | 902 | 63.0 |
| Fruit consumption | | |
| Yes 3 or more times/week | 306 | 21.4 |
| Yes 1–2 times/week | 256 | 17.9 |
| No/<1 consumption/week | 870 | 60.7 |
| Vegetables consumption | | |
| Yes 3 or more times/week | 698 | 48.7 |
| Yes 1–2 times/week | 262 | 18.3 |
| No/<1 consumption/week | 472 | 33.0 |
| Smoking (Past history) | | |
| Yes | 89 | 6.2 |
| No | 1343 | 93.8 |
| Alcohol consumption | | |
| Yes | 220 | 15.4 |
| No | 1212 | 84.6 |
| Khat Consumption | | |
| Yes | 40 | 2.8 |
| No | 1392 | 97.2 |
| BMI | | |
| <18.5 kgm2 | 211 | 14.7 |
| 18.5–24.9 kgm2 | 763 | 53.3 |
| 25.0–29.9 kgm2 | 326 | 22.8 |
| ≥30 kgm2 | 132 | 9.2 |
| Waist-to-hip ratio | | |
| Low health risk | 399 | 27.9 |
| Moderate health risk | 223 | 15.6 |
| High health risk | 810 | 56.6 |
| HIV status | | |
| HIV Positive | 53 | 3.7 |
| HIV negative | 183 | 12.8 |
| Status unknown | 1196 | 83.5 |

More than half 54.8% (CI = 52.2%-57.4%) of the study participants had multimorbidity. Of these 39.6% (n = 567) had two NCDs and 15.2% (n = 218) had three or more chronic NCDs. Of the 647 (45.2%) participants with a single morbidity, the highest proportions were for hypertension (37.9%), diabetes (34.8%) and heart disease (18.5%) (Fig 2).

The most prevalent NCDs hugely contributed to shaping the patterns of multimorbidity in this study: For example, hypertension co-existed with diabetes and heart disease in 38.2% and 19.0% of the participants, respectively. Similarly, co-occurrence of diabetes was observed among individuals with heart disease, depression and other types of reported chronic conditions. Hypertension remained the most frequently reported NCD (87.2%) among individuals living with three or more NCDs in our study, while diabetes was reported by 51% and heart disease by 39% of these participants (Table 4).

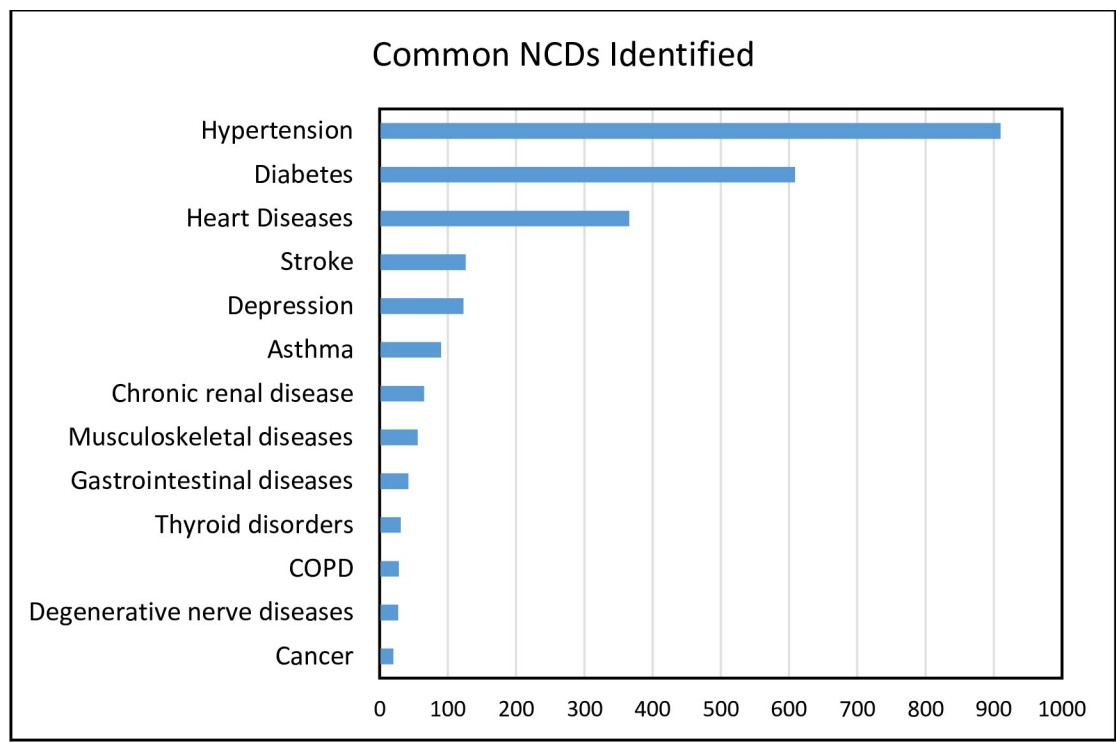

**Fig 1. List of NCDs identified and their magnitude among participants attending chronic outpatient NCDs care, Bahir Dar, Ethiopia (N = 1432).**

## Factors associated with multimorbidity of chronic NCDs

In the univariate logistic regression analysis, age, sex, residence, occupation and BMI had a statistically significant relationship with multimorbidity of chronic NCDs. However, only advanced age, being overweight and obesity were the variables that remain statistically

**Table 3. Sex distribution of NCDs among people attending chronic outpatient NCD care in Bahir Dar, Ethiopia (N = 1432).**

| Type of NCD | Males | Females | Total | P. Value |
|---|---|---|---|---|
| | Frequency (%) | Frequency (%) | Frequency (%) | |
| Hypertension | 453 (49.8) | 457 (50.2) | 910 (63.5) | 0.475 |
| Diabetes | 323 (53.0) | 286 (47.0) | 609 (42.5) | **0.010** |
| Heart disease♣ | 173 (47.3) | 193 (52.7) | 366 (25.6) | 0.467 |
| Stroke | 74 (58.7) | 52 (41.3) | 126 (8.8) | **0.025** |
| Depression | 48 (39.0) | 75 (61.0) | 123 (8.6) | **0.023** |
| Asthma | 51 (56.7) | 39 (43.3) | 90 (6.3) | 0.157 |
| Chronic renal disease | 39 (60.0) | 26 (40.0) | 65 (4.5) | 0.076 |
| Musculoskeletal diseases | 35 (62.5) | 21 (37.5) | 56 (3.9) | **0.042** |
| Gastrointestinal diseases | 22 (52.4) | 20 (47.6) | 42 (2.9) | 0.755 |
| Thyroid disorders | 2 (6.5) | 29 (93.5) | 31 (2.2) | **<0.001** |
| COPD | 10 (35.7) | 18 (64.3) | 28 (2.0) | 0.183 |
| Degenerative nerve diseases | 8 (29.6) | 19 (70.4) | 27 (1.9) | 0.051 |
| Cancer | 9 (45.0) | 11 (55.0) | 20 (1.4) | 0.823 |

♣Includes heart failure, angina, heart attack and cardiac arrhythmias

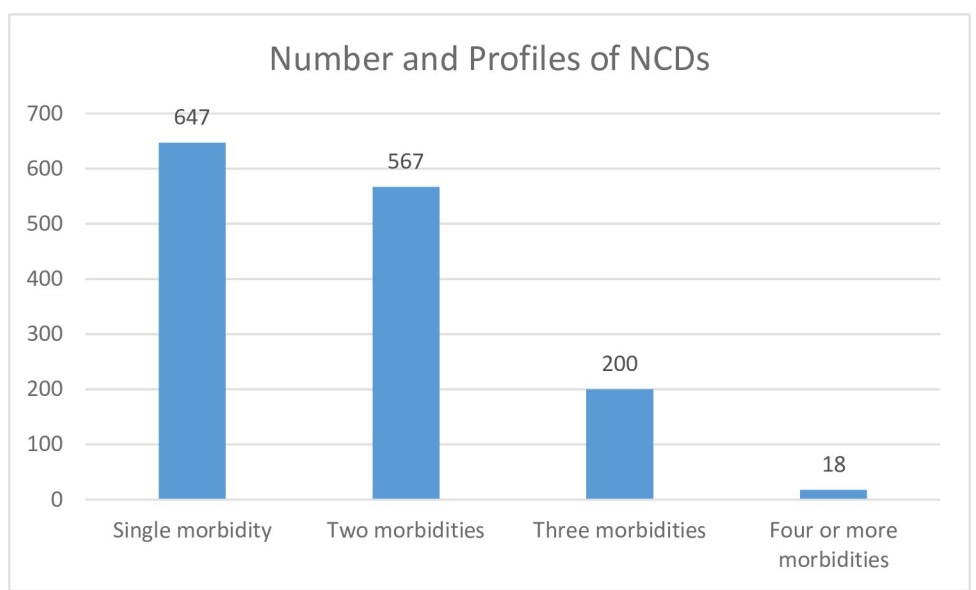

**Fig 2. Patterns of NCDs morbidity among individuals attending chronic NCDs care in Bahir Dar, Ethiopia.**

significant in the adjusted model. In our analysis, the wealth index (scored as a proxy for SES) did not show any likelihood of relationship with multimorbidity (Table 5).

Compared to participants younger than 45 years, participants falling on the age group 45–54 years (AOR: 1.5, 95%CI = 1.1, 2.1), 55–64 years (AOR: 2.5, 95%CI = 1.7, 3.5) and 65 years

**Table 4. Distribution of single NCDs and their pairwise and triples or quadruples combination, among people attending chronic outpatient NCD care in Bahir Dar, Ethiopia (N = 1432).**

| Single morbidity | | Common pairs of NCDS | | Common Triples of NCDs | |
|---|---|---|---|---|---|
| Chronic NCD | Frequency (%) | Combination | Frequency(%) | Combination | Frequency (%) |
| Hypertension alone | 245 (37.9) | Hypertension +Diabetes | 217 (38.2) | Hypertension +Diabetes+ heart disease | 19 (8.7) |
| Diabetes alone | 225 (34.8) | Hypertension + Heart disease | 108 (19.0) | Hypertension +Diabetes + depression | 18 (8.3) |
| Heart disease alone | 120 (18.5) | Hypertension + stroke | 38 (6.7) | Hypertension +Diabetes + other NCDs | 49 (22.5) |
| All other forms of single NCDs[a] | 57 (8.8) | Hypertension +Musculoskeletal diseases | 23 (4.0) | Hypertension +heart disease + other NCDs | 43 (19.7) |
| | | Hypertension + Asthma | 21 (3.7) | Hypertension + Diabetes + heart disease + other NCDs | 12 (5.5) |
| | | Hypertension + Chronic Renal diseases | 21 (3.7) | Hypertension + Diabetes + two other NCDs | 13 (6.0) |
| | | Hypertension + Depression | 18 (3.2) | Hypertension + two or other NCDs | 36 (16.5) |
| | | Hypertension + other chronic diseases | 25 (4.4) | Diabetes + two or more other NCDs | 13 (6.0) |
| | | Diabetes + Depression | 8 (1.4) | Heart disease + two or more other NCDs | 11 (5.0) |
| | | Diabetes + heart disease | 6 (1.0) | Triple or quadruple of all other NCDs | 4 (1.8) |
| | | Diabetes + other chronic NCDs | 25 (4.4) | | |
| | | Heart disease +Depression | 16 (2.8) | | |
| | | Heart disease + other chronic diseases | 27 (4.8) | | |
| | | Comorbidity of other NCDs | 14 (2.5) | | |

[a] Includes asthma, COPD, stroke, cancer and depression

**Table 5. Bivariate and multivariable analysis of the factors associated with multimorbidity, Bahir Dar, Ethiopia.**

| Variables | | | Multimorbidity | | Crude Odds Ratio (95% CI) | P Value |
|---|---|---|---|---|---|---|
| **Yes** | | | Adjusted Odds Ratio (95%CI) | P Value | **No** | |
| Age ≤44Yrs | 97 | 150 | ref | | | |
| 45-54Yrs | 199 | 200 | **1.5 (1.1, 2.1)** | **0.008** | **1.5 (1.1, 2.1)** | **0.011**** |
| 55-64Yrs | 249 | 151 | **2.5 (1.8, 3.5)** | **<0.001** | **2.6 (1.9, 3.6)** | **<0.001**** |
| 65+Yrs | 240 | 146 | **2.5 (1.8, 3.5)** | **<0.001** | **2.6 (1.9, 3.6)** | **<0.001**** |
| Sex Male | 404 | 298 | ref | | | |
| Female | 381 | 349 | **0.8 (0.6, 0.9)** | **0.042** | 0.8 (0.6, 1.1) | 0.069 |
| Residence Urban | 573 | 434 | **1.3 (1.1, 1.7)** | **0.015** | 1.1 (0.9, 1.5) | 0.356 |
| Rural | 212 | 213 | ref | | | |
| Marital status Married | 596 | 485 | ref | | | |
| Single | 189 | 162 | 0.9 (0.7, 1.2) | 0.674 | | |
| Education No formal education | 425 | 355 | ref | | | |
| Primary | 91 | 75 | 1.0 (0.7, 1.4) | 0.938 | | |
| Secondary | 98 | 73 | 1.1 (0.8, 1.6) | 0.502 | | |
| College & above | 171 | 144 | 0.9 (0.7, 1.3) | 0.952 | | |
| Occupation Employed | 168 | 160 | ref | | | |
| Farmer | 150 | 138 | 1.1 (0.7, 1.4) | 0.831 | 1.2 (0.8, 1.9) | 0.373 |
| Trader | 118 | 89 | 1.3 (0.9, 1.8) | 0.192 | 1.2 (0.8, 1.7) | 0.386 |
| Housewife | 168 | 161 | 0.9 (0.7, 1.4) | 0.965 | 1.1 (0.8, 1.6) | 0.615 |
| Retired | 98 | 43 | **2.17 (1.4, 3.3)** | **0.001** | 1.6 (0.9, 2.5) | 0.056 |
| Unemployed | 83 | 56 | 1.4 (0.9, 2.11) | 0.093 | 1.4 (0.8, 2.3) | 0.156 |
| Wealth Index Low SES | 285 | 251 | 0.8 (0.6, 1.1) | 0.133 | 0.9 (0.7, 1.3) | 0.745 |
| Middle level | 216 | 189 | 0.3 (0.6, 1.1) | 0.173 | 0.9 (0.6, 1.1) | 0.140 |
| High SES | 284 | 207 | ref | | | |
| BMI Under nutrition | 105 | 106 | 0.9 (0.7, 1.3) | 0.832 | 1.1 (0.9, 1.5) | 0.947 |
| Normal weight | 386 | 377 | ref | | | |
| Overweight | 206 | 120 | **1.7 (1.3, 2.2)** | **<0.001** | **1.6 (1.2, 2.1)** | **<0.001**** |
| Obese | 88 | 44 | **2.0 (1.4, 2.9)** | **<0.001** | **2.1 (1.4, 3.1)** | **<0.001**** |
| **Sex stratified analysis of BMI** | | | | | | |
| BMI stratified by sex (Males) | | | | | | |
| Under nutrition | 45 | 32 | 0.6 (0.3, 1.4) | 0.260 | 1.4 (0.8, 2.3) | 0.219 |
| Normal weight | 222 | 188 | ref | | | |
| Overweight | 107 | 59 | 1.5 (0.9, 2.3) | 0.099 | 1.4 (0.9, 2.0) | 0.115 |
| Obese | 30 | 19 | **2.4 (1.2, 4.7)** | **0.009** | 1.2 (0.6, 2.3) | 0.534 |
| BMI stratified sex (Females) | | | | | | |
| Under nutrition | 60 | 74 | 0.9 (0.4, 1.8) | 0.749 | 0.7 (0.5, 1.2) | 0.229 |
| Normal weight | 164 | 189 | ref | | | |
| Overweight | 99 | 61 | **2.1 (1.2, 3.6)** | **0.006** | **2.0 (1.3, 3.0)** | **0.001**** |
| Obese | 58 | 25 | **2.9 (1.6, 5.3)** | **0.001** | **3.4 (1.9, 5.9)** | **<0.001**** |

** Statistically significant in the adjusted model at 95%CI, P value ≤ 0.05. Note. The variables entered in the final model both for sex stratified and un-stratified adjusted models were the same and the significance level and parameter estimates were consistent except for the variable of interest, BMI.

or more (AOR: 2.4, 95%CI = 1.7, 3.5) had a statistically significant association with multimorbidity. Similarly, individuals classified as overweight (AOR: 1.6, 95%CI = 1.2, 2.1) or obese (AOR: 1.9, 95%CI = 1.3, 3.0) in the BMI category were 1.6 and 1.9 times more likely to have multimorbidity than those in the normal weight category, respectively. The sex stratified regression analyses revealed that the effect of obesity on the likelihood of being multimorbid was implicated by female gender. We observed that women in the overweight and obese BMI category had 2 and 3.4 times higher odds of multimorbidity compared to males with the corresponding BMI category, respectively. However, obesity in males did not show any statistically significant relationship with multimorbidity in the adjusted sex stratified logistic regression model (Table 5).

## Fitting the LCA model

One of the most important tasks in using the LCA is to accurately identify the number of underlying latent classes and correctly placing individuals into their respective classes with a high degree of confidence [37].

We used R software to run the LCA model (S1 File). Only chronic NCDs accounting for 5% or more in the prevalence estimate were eligible for fitting in the LCA model [38]. The diseases satisfying the inclusion criteria were hypertension, diabetes, heart disease, stroke, depression and asthma (scored as yes or no value). While fitting the LCA model, we employed a sequential process [36], starting with a one-class model, then adding one class at a time and continuing to run models with one additional class at a time until the best model is identified. We compared and identified the best model based on statistical and theoretical criteria.

Fit statistics, including the Akaike information criterion (AIC), Bayesian information criterion (BIC), the likelihood ratio, chi- square goodness of fit and entropy were compared and the class model with the lowest BIC was chosen as the best model [36]. The BIC is one of the most reliable fit statistic to evaluate model fit and the lower the BIC value of the statistic the more efficient the model is to fit the data [37]. In our analysis, the four class model had the lowest BIC value and the highest entropy and the addition of classes beyond four provided essentially no improvement in fit (Table 6).

Based on the average latent class posterior probability (37), the four class model classified 50.2% of the participants in class 1, 32.6% in class 2, 11.5% in class 4 and the remaining 5.7% in class 3 (Fig 3).

Upon further analysis, although not deterministic, the model predicted the probability of belongingness of a person having a "yes" response to each indicator variable. For example, a person having hypertension had a 100% probability of being assigned to class 1, 44% to class 4, 11% to class 3 and 0% to class 2. Those having diabetes had a 100% probability of being

**Table 6. Model fit statistics of each class fitted in the LCA model, Bahir Dar, Ethiopia.**

| Latent class model | AIC | BIC | Entropy | Gsq | Chisq |
|---|---|---|---|---|---|
| 1 class model | 7835.99 | 7867.58 | ———— | 688.63 | 613.86 |
| 2 class model | 7499.47 | 7567.94 | 0.80 | 338.13 | 311.90 |
| 3 class model | 7342.26 | 7447.34 | 0.72 | 166.67 | 174.05 |
| 4 class model | 7261.77 | **7402.97** | **0.86** | 71.43 | 66.69 |
| 5 class model | 7255.77 | 7434.84 | 0.67 | 52.42 | 52.43 |
| 6 class model | 7244.18 | 7460.12 | 0.70 | 26.83 | 24.62 |

Note. Gsq = the likelihood-ratio statistic; Chisq = Pearson Chi-square goodness of fit statistic; Entropy = the extent to which the groups identified are distinct from one another (its value ranges between 0 and 1 and a value ≥0.8 shows good separation of the group).

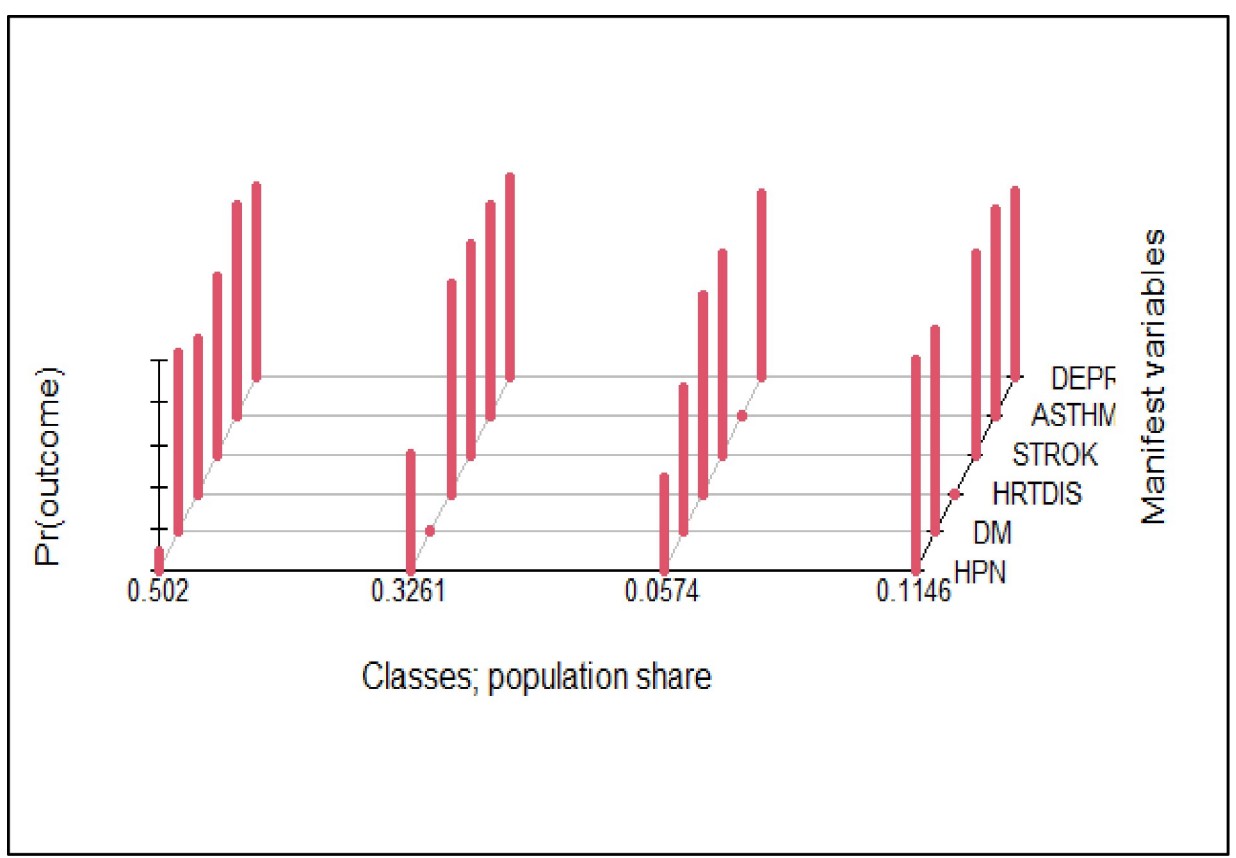

**Fig 3. Population share and distribution of conditions in the latent classes using the LCA model, Bahir Dar, Ethiopia.**

assigned to class 4, 16% to class 1, 6% to class 2 and 4% to class 3. Looking into those reported to have heart disease, the model predicted a 100% of their being in class 2, 26% in class 1 and 0% in class 3 and 4. The probability of being in class 3 for those reported to have asthma, stroke and depression was 42, 36% and 19%, respectively (Table 7).

As the observed sample is a mixture of individuals from different latent classes, individuals belonging to the same class are similar to one another such that their observed scores on a set

**Table 7. The predicted probability that a person responding "yes" to the indicator variables is being assigned in the classes in the LCA model, Bahir Dar, Ethiopia.**

| Variables | Predicted Probability | | | |
|---|---|---|---|---|
| | Latent Classes | | | |
| | **Class 1** | **Class 2** | **Class 3** | **Class 4** |
| Hypertension | 1.00 | 0.00 | 0.11 | 0.44 |
| Diabetes | 0.16 | 0.06 | 0.04 | 1.00 |
| Heart disease | 0.26 | 1.00 | 0.00 | 0.00 |
| Stroke | 0.12 | 0.06 | 0.36 | 0.01 |
| Depression | 0.09 | 0.11 | 0.19 | 0.05 |
| Asthma | 0.05 | 0.03 | 0.42 | 0.04 |
| Latent class marginal posterior probabilities | 0.52 | 0.21 | 0.05 | 0.22 |
| **Class size** | **622** | **184** | **68** | **558** |

of indicators are assumed to come from the same probability distribution [36, 37]. Hence, according to the distribution of individuals responding yes to one or more of the indicator variables, class 1 contains the majority of the individuals with hypertension, about half of the individuals with heart disease and the majority of individuals with stroke and named as a cardiovascular pattern. Class 2 contains the majority of individuals with heart disease and one third of individuals with depression and is named as a cardio-mental pattern. Class 4 contains the majority of individuals with diabetes and nearly one third of individuals with hypertension and is named as a metabolic pattern. Class 3 contains a high proportion of individuals with asthma and a small proportion of individuals with stroke and is named as a respiratory pattern. Although the class separation seems adequate and the 4 class LCA model is parsimonious, individuals are not placed adequately in a discrete and non-overlapping way. In our analysis none of the classes had a class size of less than 50 which is desirable. While class 1 includes the majority (n = 622, 43.4%), class 3 had the lowest number of individuals (n = 68, 4.7%) (Table 8).

## Predictors of latent class membership (pattern) in the LCA model

The names of the latent classes (patterns of distribution) were cardiovascular, cardio mental, respiratory and metabolic. We ran multinomial logistic regression analysis using these categories as dependent variables and by making the respiratory group as a reference category. We determined the effect of sex, age group and BMI level of individuals in predicting class membership. The model we fitted (in SPSS) satisfied the significance in the likelihood ratio of model fitness and classification accuracy criteria (predicted estimate of the group membership was higher (49.7%) than the actual group membership value (44.9%). Furthermore, as multinomial regression provides unbiased estimates when the data meet the independence of irrelevant alternatives (IIA) assumption, we ran both the fixed effect and random effect tests and checked the significance level for the Hausman test of significance) [39] using STATA. The output showed a Chi-square value of 12.59 and p-value 0.0018, indicating the rejection of the null hypothesis and the need to use the fixed effect model rather than the random effect model [39].

As shown on Table 8 below, the implication of BMI and age in dictating class membership was profound. The odds of being classified in the cardiovascular group rather than the respiratory group for individuals classified as normal weight, overweight and obese was 6.8, 20.0 and 13.8, respectively times higher than those classified in the underweight group. On the other hand, compared to individuals grouped in the respiratory class, the odds of being classified in the metabolic group were 2.8 and 3.2 times higher for individuals in the age group 55–64 years and 65+years, respectively compared to those individuals in the reference category (≤44 years). Similarly, in contrast to those classified in the respiratory class, individuals classified as overweight and obese were 4.0 and 2.1 times more likely to be classified in the metabolic group than those classified in the underweight group. Except for the metabolic group, age did not predict class membership of individuals in our analysis. Further, we did not observe any statistically significant effect of gender in predicting class membership in the multinomial logistic regression model (Table 8).

## Discussion

This study investigated the magnitude and patterns of multimorbidity, factors associated with multimorbidity and the underlying characteristics associated with an individual being in a particular latent class (cluster). We employed a blend of methods to accurately measure the burden of each individual NCD and their pairwise and triple combinations among a broad sample

**Table 8. Multinomial logistic regression analysis of the factors determining latent class membership, Bahir Dar, Ethiopia.**

| Membership | Predictor | Estimate | SE | Z | p | Odds ratio | 95% Confidence Interval Lower | Upper |
|---|---|---|---|---|---|---|---|---|
| Cardiovascular -respiratory | Intercept | -0.0575 | 0.404 | -0.1424 | 0.887 | 0.944 | 0.428 | 2.08 |
| | AGE1: | | | | | | | |
| | 45-54YRS–< = 44YRS | 0.2946 | 0.354 | 0.8316 | 0.406 | 1.343 | 0.670 | 2.69 |
| | 55-64YRS–< = 44YRS | 0.5776 | 0.384 | 1.5024 | 0.133 | 1.782 | 0.839 | 3.79 |
| | > = 65 –< = 44YRS | -0.0629 | 0.378 | -0.1662 | 0.868 | 0.939 | 0.447 | 1.97 |
| | Sex: | | | | | | | |
| | Female–Male | 0.0569 | 0.267 | 0.2130 | 0.831 | 1.059 | 0.627 | 1.79 |
| | BMI1: | | | | | | | |
| | Normal weight–Under nutrition | 1.9192 | 0.346 | 5.5516 | < .001[+] | 6.816 | 3.461 | 13.42 |
| | Overweight–Under nutrition | 2.9958 | 0.486 | 6.1627 | < .001[+] | 20.001 | 7.714 | 51.86 |
| | Obese–Under nutrition | 2.6243 | 0.594 | 4.4184 | < .001[+] | 13.795 | 4.307 | 44.19 |
| Cardiomental-respiratory | Intercept | 0.3420 | 0.407 | 0.8414 | 0.400 | 1.408 | 0.635 | 3.12 |
| | AGE1: | | | | | | | |
| | 45-54YRS–< = 44YRS | 0.0955 | 0.404 | 0.2366 | 0.813 | 1.100 | 0.499 | 2.43 |
| | 55-64YRS–< = 44YRS | 0.6876 | 0.424 | 1.6235 | 0.104 | 1.989 | 0.867 | 4.56 |
| | > = 65 –< = 44YRS | 0.7938 | 0.410 | 1.9375 | 0.053 | 2.212 | 0.991 | 4.94 |
| | Sex: | | | | | | | |
| | Female–Male | 0.5308 | 0.292 | 1.8174 | 0.069 | 1.700 | 0.959 | 3.01 |
| | BMI1: | | | | | | | |
| | Normal weight–Under nutrition | -0.0156 | 0.324 | -0.0480 | 0.962 | 0.985 | 0.522 | 1.86 |
| | Overweight–Under nutrition | -0.1386 | 0.518 | -0.2678 | 0.789 | 0.871 | 0.316 | 2.40 |
| | Obese–Under nutrition | -0.3878 | 0.665 | -0.5833 | 0.560 | 0.679 | 0.184 | 2.50 |
| Metabolic-respiratory | Intercept | 0.5859 | 0.373 | 1.5724 | 0.116 | 1.797 | 0.866 | 3.73 |
| | AGE1: | | | | | | | |
| | 45-54YRS–< = 44YRS | 0.5868 | 0.357 | 1.6443 | 0.100 | 1.798 | 0.893 | 3.62 |
| | 55-64YRS–< = 44YRS | 1.0585 | 0.385 | 2.7485 | 0.006[+] | 2.882 | 1.355 | 6.13 |
| | > = 65 –< = 44YRS | 1.1789 | 0.372 | 3.1673 | 0.002[+] | 3.251 | 1.567 | 6.74 |
| | Sex: | | | | | | | |
| | Female–Male | 0.3340 | 0.264 | 1.2666 | 0.205 | 1.396 | 0.833 | 2.34 |
| | BMI1: | | | | | | | |
| | Normal weight–Under nutrition | 1.0678 | 0.575 | 1.8577 | 0.063 | 2.909 | 0.943 | 8.97 |
| | Overweight–Under nutrition | 1.3885 | 0.459 | 3.0220 | 0.003[+] | 4.009 | 1.629 | 9.86 |
| | Obese–Under nutrition | 0.7556 | 0.303 | 2.4972 | 0.013[+] | 2.129 | 1.177 | 3.85 |

[+] Statistically significant at p-value ≤0.05

of 1432 individuals (aged 40+) attending chronic care in hospitals and specialized health facilities in Bahir Dar city, northwest Ethiopia.

The magnitude of multimorbidity estimated in this study was high (55%) with 15.2% of the participants having three or more chronic conditions. This high rate of multimorbidity indicates that multimorbidity has been and will continue to pose a huge burden on individuals, health systems and the society at large. This finding is in agreement with what is abundant in the global literature that proclaims the emergence of multimorbidity as "a norm rather than the exception" both in high-income and low-and middle-income countries [16].

Consistent with previous studies [40], hypertension remains the leading NCD affecting 65% of the people visiting health facilities in the study area, reflecting an important health

burden. Diabetes was the second highest reported chronic condition among participants and heart disease the third highest identified conditions in this patient population. These diseases played a significant role in shaping the profile of disease combinations and patterns of multimorbidity in this study. The significance of hypertension, diabetes and heart disease in dictating patterns of multimorbidity has also been reported elsewhere [9, 41, 42].

In agreement with several studies in LMICs [40, 43], the most prevalent pairwise combination was between hypertension and diabetes. Hypertension was also frequently comorbid with heart disease. This combination of NCDs in the study area may suggest that these diseases have shared risk factors, have a common pathway of occurrence or there is a causal relationship between them [36]. The pattern of combinations of conditions identified visually was almost consistent with the taxonomy assigned based on the class membership in the LCA model. This finding was consistent with studies that employed LCA models to identify patterns of multimorbidity [13]. Evidence shows the LCA model is better at accurately determining patterns of disease combinations than the traditional factor analysis models, including PCA, K-means clustering and hierarchical clustering methods [13, 44, 45].

Among the four groups of individuals in the sample, the largest disease burden was seen in the cardiovascular cluster (hypertension + heart disease), followed by the metabolic cluster (diabetes + hypertension), which together comprised 82.4% of the patterns determined. The cardio-mental and respiratory groups comprised 11.5% and 5.7% of the patterns, respectively. Previous studies have also shown that the majority of cases were clustered into either cardiovascular or metabolic clusters [10, 45]. This further substantiates the suggestion that there may be a shared pathophysiological pathways, in which the presence of one condition increases the risk of another, or due to common environmental or biological risk factors [10, 46]. The nature of coexisting medical conditions also plays an important role in predicting risk of mortality among multimorbid individuals [47].

Two of the factors (advanced age and obesity) correlated with multimorbidity have also contributed to determining latent class membership. Although it may be difficult to directly compare our findings with those of the previous studies owing to the differences in the underlying demographic characteristics, disease profiles, sources of data and study settings, the patterns of NCDs distribution and pattern of multimorbidity observed in our study concur with what is reported in some previous studies [45]. However, female sex and smoking have also contributed to class membership in previous studies [45].

## Factors associated with multimorbidity

Advanced age has always been associated with occurrences of multimorbidity globally [1, 48, 49]. Evidence shows multimorbidity starts at around 40 years of age and continues to increase in prevalence until 70 years and remains constant after then owing to the reduced survival rate of those people aged 70+ (16). In this study, we observed an increasing and statistically significant likelihood of multimorbidity among individuals aged 45+. The higher odds of multimorbidity among the people aged 55–64 and 65+ implies that longevity plays an important role in the rise in the incidence of multiple chronic conditions and their impact on quality of life and functioning [1, 16]. However, studies in high income countries reported that people living in socioeconomically deprived areas tend to develop multimorbidity 10–15 years earlier than their wealthier counterparts [1], implying that income could modify the onset of multimorbidity in low income settings.

The relationship between unhealthy diet, obesity and being overweight and multimorbidity is well established globally [48, 50, 51]. In this study, individuals classified as obese and overweight had higher odds of multimorbidity than individuals with normal weight. Globally,

obesity is a known risk factor for the occurrence of NCDs [52] and much of the increase in multimorbidity among the people living with NCDs was due to obesity [53, 54]. Studies found obese individuals, particularly women, Gen Xers and younger boomers, had a higher risk of multimorbidity than those of normal weight [55]. However, it is imperative to consider the implications of female sex in modifying the association between obesity and multimorbidity. The sex stratified analysis in our study revealed that the likelihood of living with multimorbidity was higher among women identified as overweight and obese compared to males in the same BMI category.

Although statistically insignificant, the authors found a lower likelihood of multimorbidity among females than males. This observation might be due the relative mean age difference between males (58years) and females (55years) in our study. However, some evidence shows females are more likely than males to live with multimorbidity [9], while others reported contrasting results on the association between sex and multimorbidity [48].

However, the lack of association between SES and multimorbidity in our study needs to be corroborated by future studies. The observed finding might be due the nature of data we collected to estimate wealth index using the PCA model [56].

## Implication for future research

Multimorbidity is a priority global health research agenda [57]. However, the methodologies employed to study the epidemiology of multimorbidity are not consistent globally [7, 42, 58]. A number of studies argue that estimates of multimorbidity may vary depending on the age of the population studied, the type and number of chronic conditions considered in defining multimorbidity, study settings (population based vs facility based) where multimorbidity research is conducted and the sources of data used to identify presence of chronic conditions [6, 59].

There is also a huge variability in the age group of individuals enrolled in multimorbidity studies globally [60]. The mean age of individuals who participated across most of facility based multimorbidity studies in LMICs ranged from 40–45 years [9, 40]. To facilitate comparison and because of the increased focus of studying multimorbidity among middle-aged and elderly people [3, 61], we only enrolled participants aged 40 years or more. Future studies may, however, investigate the distribution of individual NCDs and their patterns of combination in a broader age category.

Compared to population based studies, facility based studies provide a wealth of data sources to validate and triangulate information about the presence of individual NCDs and other characteristics [6]. We used two complementary methods (self-reported presence of a doctor diagnosed condition together with medical record reviews) to better estimate the magnitude of individual conditions. The chronic NCDs included in this study were selected based on a previous study [9] and analysis of preliminary data collected prior to the actual study. The magnitude of individual conditions and the multimorbidity estimated in this study were higher due to the involvement of patients on follow-up for any of the 13 selected chronic NCDs. However, the estimate in our study does not necessarily reflect the underlying epidemiology in the general population. Therefore, researchers need to be aware of these limitations when considering facility based multimorbidity studies.

The number of chronic conditions to be considered for measuring multimorbidity has been one of the areas of debate globally [6, 62]. The list of chronic condition among studies in LMICs ranged from 3–40 [9], leading to inconsistent results. Prevalence rates of 4 to 7 diagnoses or fewer than 10 chronic conditions may lead to an underestimation of the prevalence of multimorbidity [6, 63]. To better estimate multimorbidity, it is recommended that a minimum

of 12 chronic conditions, that have public health importance in a given context, be considered [6]. We used 13 of the most prevalent chronic NCDs (including those grouped together) in the study area. However, scholars in the field [36, 61] emphasized the importance of also evaluating the severity of the disease conditions beyond a simple count of combination of medical diagnoses. Others also suggest the inclusion of HIV (when prevalence is high), obesity and high cholesterol in the list to define multimorbidity [64]. However, any list of conditions used in operationalizing multimorbidity should at least consider disease burden (prevalence) and their impact on population wellbeing and survival in a given context when operationalizing the measure [57, 65].

It is argued that patients with multimorbidity are more than the sum of their individual conditions [45, 66] and that the patterns of disease combination and severity levels influence the health care to be delivered and the subsequent outcomes [66, 67]. Using robust cluster analysis techniques such as the LCA model to accurately determine patterns of multimorbidity can aid in this endeavor. The four class LCA model we fitted was parsimonious in identifying common multimorbidity patterns and the covariates determining class membership. Employing the method in the LMICs context would help replicate and compare the findings.

## Strengths and limitations of the study

Our study has the advantage of including a broad range of health care facilities providing comprehensive health services for the people living with NCDs. Guided by a published study protocol, this study employed two complementary methods to accurately identify the diagnosed chronic NCDs. The LCA model we fitted is a relatively new and plausible technique to determine the pattern of multimorbidity and identify covariates influencing class membership in a relatively efficient, reliable and valid way. However, the findings of this facility based study does not represent the underlying epidemiology and patterns of distribution of NCDs in the population. Moreover, the lack of a properly validated tools to measure patient activation, social support and multidimensional locus of control limited our ability to assess the association between these factors and multimorbidity. However, understanding the nature of people visiting health facilities, the types and magnitude of NCDs they are seeking care for and patterns of combination of NCDs among the cohort of individuals in our study would help guide future studies, and the prevention and management of multimorbidity in the country.

## Conclusions and recommendations

The magnitude of multimorbidity in this study was high. The most frequently diagnosed chronic conditions shaped the profiles of clustering and patterns of multimorbidity. Advanced age, being overweight and obesity were the factors associated with multimorbidity. Future studies may need to conduct population based studies through enrolling people from a broader age range. Further studies (such as the one we are conducting) are also needed to explore the effect of multimorbidity on quality of life, functioning and survival, and to assess how health services are oriented and organized to meet the care needs of the people living with multiple chronic conditions in the country.

## Supporting information

**S1 File. R script fitted for the LCA model.**
(SAV)

**S2 File.**
(RHISTORY)

## Acknowledgments

We would like to thank Bahir Dar University, data collectors, supervisors, facilities leaders and study participants for their support in making this study a reality.

## Author Contributions

**Conceptualization:** Fantu Abebe Eyowas, Marguerite Schneider, Shitaye Alemu, Fentie Ambaw Getahun.

**Data curation:** Fantu Abebe Eyowas.

**Formal analysis:** Fantu Abebe Eyowas.

**Funding acquisition:** Fantu Abebe Eyowas.

**Investigation:** Fantu Abebe Eyowas.

**Methodology:** Fantu Abebe Eyowas, Marguerite Schneider, Shitaye Alemu, Sanghamitra Pati, Fentie Ambaw Getahun.

**Resources:** Fantu Abebe Eyowas.

**Supervision:** Fantu Abebe Eyowas, Marguerite Schneider, Fentie Ambaw Getahun.

**Validation:** Fantu Abebe Eyowas, Marguerite Schneider.

**Writing – original draft:** Fantu Abebe Eyowas.

**Writing – review & editing:** Fantu Abebe Eyowas, Marguerite Schneider, Shitaye Alemu, Sanghamitra Pati, Fentie Ambaw Getahun.

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
