## [Decision Letter · Decision Letter 0]

12 Jan 2022

PONE-D-21-32202Magnitude, pattern and correlates of multimorbidity among patients attending chronic outpatient medical care in Bahir Dar, northwest Ethiopia: the application of latent class analysis modelPLOS ONE

Dear Dr. Eyowas,

Thank you for submitting your manuscript to PLOS ONE. After careful consideration, we feel that it has merit but does not fully meet PLOS ONE’s publication criteria as it currently stands. Therefore, we invite you to submit a revised version of the manuscript that addresses the points raised during the review process.

The manuscript has been assessed by two reviewers. Their comments are appended below. The reviewers have raised some of major concerns about the manuscript, and in particular they feel that important methodological issues exist that affect the technical soundness of your study, and the conclusions of the paper. Notwithstanding, we are returning the paper to authors as major revision decision. We request authors to conduct a detailed review of the comments to enable the manuscript to be able for publication after future reviews.

We look forward to receiving your revised manuscript.

Kind regards,

Bruno Pereira Nunes, Ph.D.

Academic Editor

PLOS ONE

Journal Requirements:

Reviewers' comments:

Reviewer's Responses to Questions

**Comments to the Author**

1. Is the manuscript technically sound, and do the data support the conclusions?

Reviewer #1: Yes

Reviewer #2: No

2. Has the statistical analysis been performed appropriately and rigorously? 

Reviewer #1: Yes

Reviewer #2: No

3. Have the authors made all data underlying the findings in their manuscript fully available?

Reviewer #1: Yes

Reviewer #2: Yes

4. Is the manuscript presented in an intelligible fashion and written in standard English?

Reviewer #1: Yes

Reviewer #2: Yes

5. Review Comments to the Author

Reviewer #1: This papers investigates the magnitute, pattern and associated factors of multimorbidity. It presents a well-detailed methodology and its discussion is extensive. It is important, if possible, to share R scripts for the LCA model, in order to make it more transparent and even reproducible. It is important for authors to cite and compare papers that used LCA for similar objectives.

Reviewer #2: Congratulations for this interesting article. It presents an impressive effort to explore MM in a population of adult ambulatory patients from Ethiopia. However, there are some major issues that the authors must address in order to support the results they are presenting here.

INTRO - very good review of the main issues regarding research in MM and health care of patients with MM. However, readers of this article will most likely know the whole picture. I'd suggest you try to summarise the information, focusing on the topics covered by the study. The last paragraph makes sense in a thesis as a justification, not so much here.

METHODS - the study protocol is well described but what was the main purpose of this study? Reading the description one can realise that this article is just one part of the whole project. Stating the aim of the whole project in the first paragraph will help the reader understand the differences. "Patient activation", "Social networking and support system", "Multidimensional

227 health locus of control", and "Wealth Index" are all multidimensional scales. However, none of these scales, nor their references, present documentation showing their adaptation to the context where the study took place. One of them shows validation in a Hungarian population. Do you have studies presenting that validation? This is a main issue to the study and if they were not validated, it hinders the quality of data collected. Row 257 informs that the questionnaires were translated according standards, however, it is not enough to validate the instruments. Also, it is not well explained how info about dietary habits, behavioral and lifestyle patterns was collected. What were the instruments used here?

RESULTS - info about Lifestyle, behavioral and psychosocial characteristics needs to be supported by well documented instruments. In row 384 the information about the association between age and MM is well written. In the abstract the same words should be used, instead of "The likelihood of developing multimorbidity was higher among". Since it is a cross-sectional study it is hard to support this claim. Same thing in row 388 " the effect of obesity on the likelihood of developing multimorbidity" and it must be changed. Use likelihhod of having MM or being multimorbid.

DISCUSSION - first paragraph can be supressed. Rows 548 to 566 lack the support of validated instruments, as mentioned before. Rows 567 to 581 does not relate to the study. They revisit information presented in the introduction but with no new evidence from the study. Same for rows 582 to 598.

Strengthen and Limitations - although I agree that a facility-baased research is key to bring evidences to healthcare systems to support patient centered care, the lack of proper instruments to collect information is crucial.

I'll add another limitation of the study: reading all the manuscript I can see the amount of effort you must have dedicated to this research. It started from a study hypothesis and it should be described in the beginning of the manuscript.

CONCLUSION - it will change with all the suggestions from the reviewers but I strongly recomend that you keep it in one sole paragraph, not four. Reading it sound almost alike to the abstract and intro sections.

My final suggestions - You have enough material to produce two or even three articles. If you cut the part related to the instruments that were not validated to your context and keep the analysis as they were presented only with age, sex and obesity, it is already a very good article. However, using the instruments brought a lot of noise to your analysis and probably the lack of validation is the reason why you didn't get significance for these variables.

Keep it short and resubmit a shorter version, presenting only the main info you already have and can support (age, sex, obesity and morbidities).

With no more words to say, congratulations for your work.

6. PLOS authors have the option to publish the peer review history of their article (what does this mean?). If published, this will include your full peer review and any attached files.

Reviewer #1: No

Reviewer #2: **Yes: **ADELSON GUARACI JANTSCH

---

## [Author Response · Author response to Decision Letter 0]

8 Feb 2022

Rebuttal Letter: Point-by-point responses to each of the comments given by the Academic editor and reviewers

PONE-D-21-32202

Manuscript title: Magnitude, pattern and correlates of multimorbidity among patients attending chronic outpatient medical care in Bahir Dar, northwest Ethiopia: the application of latent class analysis model

PLOS ONE

Academic Editor’s comments 

Thank you for submitting your manuscript to PLOS ONE. After careful consideration, we feel that it has merit but does not fully meet PLOS ONE’s publication criteria as it currently stands. Therefore, we invite you to submit a revised version of the manuscript that addresses the points raised during the review process.

Authors response/s 

We are thankful for the opportunity given to us to revise and submit the manuscript according to the comments and suggestions provided. We have provided a point by point response in this document and we have used track changes in the main manuscript. Our responses are highlighted in blue, and those highlighted in red are to show hoe the revisions have been made in the manuscript. 

Academic Editor’s comments 

The manuscript has been assessed by two reviewers. Their comments are appended below. The reviewers have raised some of major concerns about the manuscript, and in particular they feel that important methodological issues exist that affect the technical soundness of your study, and the conclusions of the paper. Notwithstanding, we are returning the paper to authors as major revision decision. We request authors to conduct a detailed review of the comments to enable the manuscript to be able for publication after future reviews.

Authors response/s 

Thank you again! We have carefully read all the comments and suggestions given and addressed all of them. We revised the manuscript to meet PLOS ONE's style requirements. We have also thoroughly copyedited the manuscript for language usage, spelling, and grammar. One of the co-authors, Marguerite Schneider (native English speaker) conducted in the proofreading and editing of the manuscript.

Review Comments to the Author

Reviewer #1: 

This papers investigates the magnitute, pattern and associated factors of multimorbidity. It presents a well-detailed methodology and its discussion is extensive. It is important, if possible, to share R scripts for the LCA model, in order to make it more transparent and even reproducible. It is important for authors to cite and compare papers that used LCA for similar objectives.

Authors response/s 

Thank you for giving us these valuable comments and suggestions. Now, the R scripts we ran for the LCA model are submitted as supplementary file (S 4 and 5).

We have cited more articles that used LCA and compared the findings with our observation. The revisions made are indicated on page 20, lines 937-940. 

Reviewer #2: 

Comment 1

Congratulations for this interesting article. It presents an impressive effort to explore MM in a population of adult ambulatory patients from Ethiopia. However, there are some major issues that the authors must address in order to support the results they are presenting here.

Authors response/s 

Thank you for the critical review and provision of valuable feedback and comments. We have duly addressed all the comments and recommendations. The page and line numbers where changes/revisions have been made in the main manuscript are indicated under each of the reviewer/s comments described below.

Comment 2

INTRO - very good review of the main issues regarding research in MM and health care of patients with MM. However, readers of this article will most likely know the whole picture. I'd suggest you try to summarise the information, focusing on the topics covered by the study. The last paragraph makes sense in a thesis as a justification, not so much here.

Authors response/s 

Thank you for the comments. We have summarized the introduction section and included only the content that is particularly relevant for the current study. The revisions made are on page 3, line 125-137, 138-149 and line 249-253. 

Comment 3

METHODS - the study protocol is well described but what was the main purpose of this study? Reading the description one can realise that this article is just one part of the whole project. Stating the aim of the whole project in the first paragraph will help the reader understand the differences. "Patient activation", "Social networking and support system", "Multidimensional

227 health locus of control", and "Wealth Index" are all multidimensional scales. However, none of these scales, nor their references, present documentation showing their adaptation to the context where the study took place. One of them shows validation in a Hungarian population. Do you have studies presenting that validation? This is a main issue to the study and if they were not validated, it hinders the quality of data collected. Row 257 informs that the questionnaires were translated according standards, however, it is not enough to validate the instruments. Also, it is not well explained how info about dietary habits, behavioral and lifestyle patterns was collected. What were the instruments used here?

Authors response/s 

Thank you very much for critically reviewing the method section and providing relevant and educational comments. To give insight about how the current study is related to the ongoing research project, we added the paragraph below in the introduction. Page 3, line 249-253 that we explained this as follows.

“This study is part of an ongoing longitudinal study being conducted in a broader range of health care facilities providing health care for the people living with NCDs in the region. This study aimed to determine the magnitude and patterns of multimorbidity and associated factors among individuals attending chronic outpatient medical care in public and private health facilities in Bahir Dar city, Ethiopia.” 

We have also elaborated on this in the first part of the method section on Page 3, line 255-257.

Comment 3.1

"Patient activation", "Social networking and support system", "Multidimensional

227 health locus of control", and "Wealth Index" are all multidimensional scales. However, none of these scales, nor their references, present documentation showing their adaptation to the context where the study took place. One of them shows validation in a Hungarian population. Do you have studies presenting that validation? This is a main issue to the study and if they were not validated, it hinders the quality of data collected. Row 257 informs that the questionnaires were translated according standards, however, it is not enough to validate the instruments.

Authors response/s 

As said by the reviewer, the tools used to collect data on patient activation, social networking and support systems, and locus of control were not adapted and validated in the context where this study was conducted. However, the tools were used after translation into the local language (Amharic) and pilot testing using 2% of the sample (n=29) in one public hospital and one private hospital which were not involved in the main study. However, we acknowledge that translation and pilot testing are not sufficient to ensure validity of the instruments and removed all the related data for these instruments from our final analysis. The content describing these variables in the introduction, methods, results and discussion has also been removed. The limitations in this regard are presented on page 22, line 1130-1132. “Moreover, the lack of a properly validated tools to measure patient activation, social support and multidimensional locus of control in the context limited our ability to assess the association between these factors and multimorbidity.”

However, given that the wealth index was assessed using a tool that has long been validated and used in Ethiopia, we described this in detail and retained the concept and included the findings accordingly in the paper. The text added is as follows on page 6, line 392-395: “Wealth Index (a latent construct) at household level was generated from a combination of material assets and housing characteristics according to the demography and health survey guideline used in Ethiopia” We re-ran the analyses after eliminating these factors and reported the revised results accordingly. 

Comment 3.2

Also, it is not well explained how info about dietary habits, behavioral and lifestyle patterns was collected. What were the instruments used here?

Authors response/s 

We have explained how information about dietary habits, behavioral and lifestyle patterns was collected. Please check on page 6, line 363-379, page 9, line 617-619. We explained this in the main manuscript as follows; “We used the WHO’s STEPs survey tool (26) which was adapted and validated in Ethiopia to assess dietary habits [amount and frequency of fruit and vegetables consumption and amount of daily salt consumption], and behavioral and lifestyle patterns [alcohol consumption, smoking, Khat consumption and physical exercise].”

Comment 4

RESULTS - info about Lifestyle, behavioral and psychosocial characteristics needs to be supported by well documented instruments. 

Authors response/s 

Thank you for the important comments given on this section 

As explained above, the methods and tools we used to assess lifestyle, and behavioral and dietary practices are in line with the recommendations indicated in the nationally adapted STEPs survey guideline, substantiating that the tools and procedures we employed are valid and replicable. 

Comment 4.1

In row 384 the information about the association between age and MM is well written. In the abstract the same words should be used, instead of "The likelihood of developing multimorbidity was higher among". Since it is a cross-sectional study it is hard to support this claim. Same thing in row 388 " the effect of obesity on the likelihood of developing multimorbidity" and it must be changed. Use likelihhod of having MM or being multimorbid.

Authors response/s 

Thank you for your feedback and recommendations. We have revised the statements accordingly. Please check on page 2, line 44 on the abstract section and page 13, line 770. 

Comment 5

DISCUSSION - first paragraph can be supressed. Rows 548 to 566 lack the support of validated instruments, as mentioned before. Rows 567 to 581 does not relate to the study. They revisit information presented in the introduction but with no new evidence from the study. Same for rows 582 to 598.

Authors response/s 

Thank you for these comments too. We have carefully revised the discussion. We deleted the first paragraph. Please check on page 19, line 903-908. We removed the section that was describing locus of control. Page 21. 

Comment 5.1 

Rows 567 to 581 does not relate to the study. 

Authors response/s 

Thank you! We removed the section/s that were not directly related to the current study. Please check the discussion.

Comment 5.2 

They revisit information presented in the introduction but with no new evidence from the study. Same for rows 582 to 598.

Authors response/s 

Thank you! We revised the contents that was not supported by the findings in this study. Please check the discussion on page 21.

Comment 6

Strengthen and Limitations - although I agree that a facility-baased research is key to bring evidences to healthcare systems to support patient centered care, the lack of proper instruments to collect information is crucial.

Authors response/s 

Thank you!

Although we tried to improve the quality of the multidimensional tools used the first time in the country through checking internal consistency, cognitive interview, pilot testing and translation, we agreed to remove contents and results referring to LoC, social support and networking and patient activation. 

We explained in the limitations section that we were unable to report the results on patient activation, social support and networking and LoC as the tools we used were not fully adapted and validated in the study context. Please check on page 22, line 1143-1145 under the limitations section. However, the tools we used to measure wealth index, dietary, lifestyle and behavioral characteristics have been very well adapted and used in the country. This is explained above (under the comments on the methods section).

Comment 6.1

I'll add another limitation of the study: reading all the manuscript I can see the amount of effort you must have dedicated to this research. It started from a study hypothesis and it should be described in the beginning of the manuscript.

Authors response/s 

Thank you again for the comment. We have described the nature of the entire study project in the introduction and methods sections. We hope readers would also review the study protocol we published to guide the present and future studies. 

Comment 7

CONCLUSION - it will change with all the suggestions from the reviewers but I strongly recomend that you keep it in one sole paragraph, not four. Reading it sound almost alike to the abstract and intro sections.

Authors response/s 

Thank you again for the comment.

We have reduced the content of the conclusion and made only one paragraph as suggested.

Comment 8

My final suggestions - You have enough material to produce two or even three articles. If you cut the part related to the instruments that were not validated to your context and keep the analysis as they were presented only with age, sex and obesity, it is already a very good article. However, using the instruments brought a lot of noise to your analysis and probably the lack of validation is the reason why you didn't get significance for these variables.

Authors response/s 

Thank you for these important observations and suggestions.

We have revised the entire manuscript to only focus on the findings that were supported by the data collected by validated tools.

Keep it short and resubmit a shorter version, presenting only the main info you already have and can support (age, sex, obesity and morbidities).

Authors response/s 

Thank you so much. We have carefully revised the content of the manuscript and made it focused and succinct. 

With no more words to say, congratulations for your work.

Authors response/s 

We are grateful for all of the comments given.

---

## [Decision Letter · Decision Letter 1]

5 Apr 2022

Magnitude, pattern and correlates of multimorbidity among patients attending chronic outpatient medical care in Bahir Dar, northwest Ethiopia: the application of latent class analysis model

PONE-D-21-32202R1

Dear Dr. Eyowas,

We’re pleased to inform you that your manuscript has been judged scientifically suitable for publication and will be formally accepted for publication once it meets all outstanding technical requirements.

Kind regards,

Bruno Pereira Nunes, Ph.D.

Academic Editor

PLOS ONE

Additional Editor Comments (optional):

Reviewers' comments:

Reviewer's Responses to Questions

**Comments to the Author**

1. If the authors have adequately addressed your comments raised in a previous round of review and you feel that this manuscript is now acceptable for publication, you may indicate that here to bypass the “Comments to the Author” section, enter your conflict of interest statement in the “Confidential to Editor” section, and submit your "Accept" recommendation.

Reviewer #2: All comments have been addressed

2. Is the manuscript technically sound, and do the data support the conclusions?

Reviewer #2: Yes

3. Has the statistical analysis been performed appropriately and rigorously? 

Reviewer #2: Yes

4. Have the authors made all data underlying the findings in their manuscript fully available?

Reviewer #2: Yes

5. Is the manuscript presented in an intelligible fashion and written in standard English?

Reviewer #2: Yes

6. Review Comments to the Author

Reviewer #2: Dear authors, congratulations! The article has improved substantialy and it is now ready to be published. Congratulations for performing this very interesting study in a low resource setting. We need more studies exploring chronic health conditions and multimorbidity in LMIc and settings outside Europe and North America.

One final detail you must fix. please, add in table 3 what the p-value refers to. I assume it is a chisquare test but you must add this info to the table.

One last time, congratulations!

7. PLOS authors have the option to publish the peer review history of their article (what does this mean?). If published, this will include your full peer review and any attached files.

Reviewer #2: **Yes: **ADELSON GUARACI JANTSCH

---

## [Editor Report · Acceptance letter]

14 Apr 2022

PONE-D-21-32202R1 

Magnitude, pattern and correlates of multimorbidity among patients attending chronic outpatient medical care in Bahir Dar, northwest Ethiopia: the application of latent class analysis model 

Dear Dr. Eyowas:

I'm pleased to inform you that your manuscript has been deemed suitable for publication in PLOS ONE. Congratulations! Your manuscript is now with our production department. 

Kind regards, 

on behalf of

Dr. Bruno Pereira Nunes 

Academic Editor

PLOS ONE